# Delays in TB Diagnosis and Treatment Initiation in Burkina Faso during the COVID-19 Pandemic

**DOI:** 10.3390/tropicalmed7090237

**Published:** 2022-09-09

**Authors:** Adama Diallo, Adjima Combary, Vanessa Veronese, Désiré Lucien Dahourou, Salifou Ouédraogo, Isidore Tiandiogo Traoré, Corinne Simone Merle

**Affiliations:** 1National Tuberculosis Control Program, Ouagadougou P.O. Box 6632, Burkina Faso; 2Special Programme for Research & Training in Tropical Diseases (TDR), World Health Organization, 1211 Geneva, Switzerland; 3Département Biomédical/Santé Publique, Institut de Recherche en Sciences de la Santé, Ouagadougou P.O. Box 7192, Burkina Faso; 4Institut Supérieur des Sciences de la Santé, Université Nazi Boni, Bobo-Dioulasso P.O. Box 1091, Burkina Faso

**Keywords:** tuberculosis, COVID-19, treatment delays, Burkina Faso

## Abstract

The COVID-19 pandemic has significantly disrupted TB services, particularly in low resource settings. In Burkina Faso, a cross-sectional ‘before and after’ study was conducted to assess the impact of COVID-19 on access to TB services. Data was collected in two phases (Phase 1: December 2017–March 2018, and 2: October–December 2020) to estimate and compare various patient and system delays among TB patients before and during COVID-19 and explore changes in treatment seeking behaviors and practices. 331 TB patients were recruited across the two phases. A significant increase in median time between first symptom and contact with TB service (45 days vs. 26 days; *p* < 0.01) and decrease in median time between first contact and diagnosis, and treatment initiation, respectively, during COVID-19 compared to before. Fewer patients reported using public health centers and more patients reporting using private facilities as the point of first contact following TB symptom onset during the COVID-19 period compared to before. These findings suggest that COVID-19 has created barriers to TB service access and health seeking among symptomatic individuals, yet also led to some efficiencies in TB diagnostic and treatment services. Our findings can be help target efforts along specific points of the TB patient pathway to minimize the overall disruption of COVID-19 and future public health emergencies on TB control in Burkina Faso.

## 1. Introduction

Tuberculosis (TB) remains a serious public health problem in Burkina Faso. In 2019, the incidence of TB was estimated at 47 cases per 100,000 population and mortality at 9.7 cases per 100,000 population [1]. The national response to TB has been led by the National TB Control Programme (PNT) since 1995, under the Ministry of Health. Despite the availability of free diagnosis and treatment of TB, national TB control priorities in Burkina Faso have not yet been achieved [2]. Compared to latest national estimates of 47 cases per 100,000, the TB notification rate over the past five years has remained stable at below 30 cases per 100,000, suggesting ongoing issues with under-detection and/or under-reporting of the incidence TB in the country. Similarly, TB mortality rate has remained high, ranging between 10.3% in 2016 and 8.3% in 2019 [2]. Late or missed diagnoses of TB contributes to delays in treatment initiation, higher rates of morbidity, mortality and sustains ongoing community transmission [3,4,5,6,7,8,9].

The global COVID-19 pandemic has exacerbated many of the challenges of TB control, particularly in low- and middle-income countries [10]. The first case of COVID-19 in Burkina Faso was confirmed on 9 March 2020, and as of June 2022, more than 21,000 cases have been confirmed [11]. Faced with the pandemic, a national response was organized by the government to control the circulation of SARS-COV-2, which included: (i) quarantine and imposition of a curfew for cities where a case of COVID-19 has been detected; (ii) isolation and care of individuals with suspected and confirmed COVID-19; (iii) the development and dissemination of awareness-raising messages on COVID-19 risk reduction and public safety; (iv) the suspension of urban public transport modalities, closure of national borders and travel bans, and; (v) closure of mass gathering places, such as markets, places of worship, and educational facilities [12]. The national response to COVID-19 has also seen the reallocation of health workers from other sectors to the national response.

The COVID-19 pandemic and the resulting restrictions aimed at curbing its spread, has caused much disruption to public life and to the health system more broadly. The pandemic has had a profound impact on TB services’ efforts in prevention, detection and case management, particularly in resource-limited settings [13,14] A recent analysis by WHO found an 18% decrease in TB case detection during COVID-19 [15], pointing to barriers to access or delivery of key TB services during the pandemic, and highlighting the potential of COVID-19 to recent gains in TB control.

This study aimed to assess the impact of the COVID-19 pandemic on access to timely TB diagnosis and treatment initiation among TB patients in Burkina Faso, in order to inform appropriate mitigation strategies.

## 2. Materials and Methods

### 2.1. Study Design

A quantitative, cross-sectional “before-and-after” study was conducted among TB patients in central eastern and central northern regions of Burkina Faso to explore the impact of the COVID-19 pandemic on TB diagnosis and treatment initiation and changes to health seeking behavior and practices.

### 2.2. Settings

This study was conducted in the central eastern and central northern regions of Burkina Faso, home to approximately 16% of the national population, with 1,704,810 and 1,787,082 habitants, respectively [16]. TB notifications from these two regions represented 13.4% of all cases reported nationally in 2020, and 19% of TB-related deaths.

The health care system in both regions is organized in two levels: the primary level, which is composed of health and medical centers, and the secondary level, which is represented by medical centers with surgical capacities and serve as reference hospitals for the districts. TB screening is typically conducted at the primary level, diagnostic and treatment centers (CDT). Each region has seven CDTs in each region, including one that is attached to the central regional hospital and equipped with GeneXpert. Upon suspicion of TB, sputum samples are collected and sent to nearest CDT for processing by the attached laboratory. Bacteriological sputum examination is typically used as the first-line diagnostic test while GeneXpert testing is reserved for the diagnosis of rifampicin resistance in previously-treated TB patients, and as a first-line diagnostic test for certain high-risk populations, such as people living with HIV, children, prisoners, gold washers, diabetics and the elderly. For the population served by the CDT attached to the central regional hospital, GeneXpert is used as the first-line diagnostic test [17].

### 2.3. Study Outcomes

In order to assess the impact of the COVID-19 pandemic on timely access to TB diagnosis and treatment initiation and health seeking behaviors and practices, the length of time in days between specified events was calculated and used to identify the following types of delays: (1) patient delay, defined as the time between the first symptoms reported by the patient and the first consultation at a health center; (2) system delay, defined as the time between the first consultation in a health center and the initiation of anti-TB treatment; (3) overall delay, defined as the time between the onset of symptoms and the initiation of anti-TB treatment. This delay is the sum of the patient and system delay; (4) diagnostic delay, defined as the time between the first consultation at a health center and the diagnosis of TB and; (5) processing time, defined as the time between the diagnosis of TB and the initiation of anti-TB treatment.

### 2.4. Data Collection and Study Variables

Data collection took place in two phases: the ‘before’ phase occurred between December 2017 and March 2018 and the ‘during’ phase occurred between October and December 2020, eight months after the first case of COVID-19 was notified in Burkina Faso. The ‘before’ phase data was collected as part of another study on the access to TB services in Burkina Faso and was used in this study as secondary data to provide a comparison to the data collected during the COVID-19 pandemic.

In both periods, a standardized, quantitative questionnaire was used to assess the access and time to TB diagnosis and treatment among TB patients in Burkina Faso. The questionnaire was administered to eligible TB patients by the health worker responsible for TB treatment in each of the 14 CDT study sites, either in French or in the patient’s own language. The questionnaire collected data on the following variables: socio-demographic status (age, sex, profession, place of residence, marital status), clinical presentation (weight, height, onset of symptoms, type of symptoms), comorbidities (diabetes, alcohol consumption habits, smoking status), outcomes (microscopy test results, HIV status) and therapeutic data (seeking care, initiation of treatment, treatment regimen, treatment interruption), therapeutic itinerary (pathways to public and/or private TB services, to traditional practitioner, self-medication).

### 2.5. Participants and Sample Size Calculations

Eligible participants were defined as any individual (no age limitations) with bacteriologically confirmed or clinically diagnosed TB, who were either newly diagnosed or currently enrolled at one of the 14 CDTs in the two regions during the ‘before’ or ‘during’ study period, and who provided consent to participate in the study. Individuals who did not reside in the health areas of the CDT study sites were excluded from the study.

Consecutive sampling was used to invite all eligible participants into the study during the two study periods. A sample size of approximately 320 was set for the ‘during’ COVID phase to match the sample size of the original study, giving a combined sample size of approximately 600. Recruitment was conducted by the health workers responsible for TB treatment in each of the participating CDTs.

### 2.6. Data Analysis

Quantitative data from the two periods were merged into a single data set for cleaning and analysis. Responses were used to calculate the duration in days between various events of interest across the specified delays (as described above), and descriptive statistics (mean and standard deviation for variables that were normally distributed, and median and interquartile range for non-normally distributed variables) were used to characterize the dataset. Student’s T test and Wilcoxon and/or Kruskal–Wallis tests were to identify significant differences in mean and medians differences in duration of delays, respectively, between the two periods across socio-demographic and clinical variables. A degree of significance *p* < 0.05 was considered statistically significant. All analysis was performed using R version 3.6.2 [18].

### 2.7. Ethics

The study was approved by the Burkina Faso national committee of ethics for research in health (deliberation number N° 2020-9-194). Patient data were collected in this study with strict respect for confidentiality and anonymity. To preserve anonymity, the patient treatment number (TB register number) was used instead of the name. This anonymity was preserved during data entry and analysis.

## 3. Results

### 3.1. Participant Characteristics

A total of 331 TB patients participated in the study (167 in the ‘before COVID’ period, and 164 in the ‘during COVID’). Overall, the majority of patients identified as male (74.6%), married (58.6%), employed as a farmer/livestock breeder (36.3%), resided in the central east region (57.1%). The average age was 38.7 years. There were no significant differences found in regard to sex, age, or region between participants recruited for the first and second phase, respectively. A significant difference was found between the proportion of participants identifying as single, or as a farmer/livestock breeder between the two phases (Table 1).

Regarding the clinical profile of participants, approximately one in three identified as a current or former tobacco smoker (33.8%), one in four (26.3%) reported consuming alcohol, and few identified as having diabetes (1.2%) or HIV (5.7%). The majority of participants were categorized as underweight (BMI < 18.5; 56.2%) with a mean BMI of 18.3. Significant differences were identified in mean BMI and proportion of participants categorized as normal BMI between the two periods; all other variables remained consistent (Table 2).

Regarding TB symptoms, the majority of participants reported cough (91.5%), fever (80%), fatigue (64.3%), lack of appetite (63%), and weight loss (78%) at the time of survey. Approximately one in three reported sweats (34.3%) and 5% reported pallor. Three quarters of participants had bacteriologically diagnosed TB, while the rest were clinically diagnosed (75.8 and 24.2% respectively). Significantly fewer participants reported cough or fever in phase two compared to phase one, while a significantly higher proportion of participants were clinically diagnosed in phase two (Table 2).

### 3.2. Access Pathways to TB Services among Recruited TB Patients before and during COVID-19

Figure 1 shows the pathways to TB care services among participants following the onset of symptoms during the two periods. Before COVID, the majority of patients (56.3%) reported public health centers as the first point of contact following the onset of symptoms. This proportion was significantly lower during the COVID period, with 44.5% of TB patients reporting public health centers as the first point of contact following symptoms.

Conversely, there was a marked increase in the proportion of participants reporting use of private health centers as first point of contact during COVID compared to before, from 4.8% to 17.1%, respectively (*p* = 0.003). During the two periods, there was also a slight decrease in the proportion of participants reporting traditional practitioners as first point of contact (18.2% vs. 20.9%; *p* = 0.54) following symptom onset, and an increase in of the proportion of participants reporting self-medication as their first response (20.1% vs. 18%; *p* = 0.62).

Participants were asked about their secondary pathways following their first action after the onset of symptoms. Among all patients reporting public health centers as their first point of contact, there was a significant increase in the proportion of patients reporting secondary visits to private clinics during COVID compared to before COVID (33% vs. 4% respectively; *p* < 0.001).

The proportion of patients who had secondarily resorted to self-medication did not significantly change across the two period during COVID-19 compared to the before COVID-19 period (15% during vs. 11% before COVID; *p* = 0.46). There was also no significant change in the proportion of patients who went to the traditional practitioner after the public health centers between the two periods (18% during vs. 21% before COVID; *p* = 0.69).

### 3.3. Delays in Health Care Access before and during COVID-19

Across both study periods, the median overall delay (the time between the onset of symptoms and the initiation of anti-TB treatment) was 67 days. The median for specific delays were 31 days for patient delay (time elapsed between the date of onset of the first symptoms and the date of contact with a modern health structure); 33 days for system delay (the time elapsed between the date of the first contact with the healthcare system, such as health center, hospital, etc., and the date of initiation of anti-TB treatment); 29.5 days for median diagnostic time (the time elapsed between the date of first contact with the health care system and the date of diagnosis of TB); and two days for processing time (Table 3).

There was a significant increase in the median patient delays between the ‘before’ and ‘during’ COVID periods (45 days vs. 26 days; *p* < 0.01) and the median number of days for processing delays (2 days vs. 1 day; *p* < 0.01). Conversely, there was a significant decrease in the median number of days for system delays (20 days vs. 44 days; *p* < 0.01), diagnostic delays (17 days vs. 40 days; *p* < 0.01), and overall delay (55 days vs. 90.5 days; *p* < 0.01) in the COVID-19 period compared to before the pandemic (Table 3).

## 4. Discussion

In this study, we estimated the overall time between symptom onset and health seeking leading to the initiation of anti-TB treatment among patients in Burkina Faso before and during the COVID-19 pandemic. We found significant increases in the amount of time between first onset of symptom and attendance at a health facility by patients across the two time periods, yet conversely, found that the overall length of time between symptom onset and treatment initiation significantly decreased during COVID-19, suggesting perhaps unintended efficiencies in the health system due to decreased demand and health seeking behaviors. Collectively, these findings can be used to help target efforts along specific points of the TB patient pathway to minimize the overall disruption of the COVID-19 pandemic and other future public health emergencies on national efforts to control TB in Burkina Faso.

Globally, there has been a large decline in TB notifications during the COVID pandemic, which dropped by 18% between 2019 and 2020, reversing many years of progress against TB [15]. There are many possible contributors to the decreased number of case notifications, including changes in the willingness or ability of TB patients to seek care. In our study, we found a significant increase in the median number of days between symptom onset and attendance at a health facility, which almost doubled during the COVID-19 period, to a median of 45 days. Much of the current literature suggests that fear of exposure to COVID-19 has created a significant barrier to health care seeking across both high- and low-income settings around the world [19,20,21] and may be one factor behind the increased delay in health seeking observed in our study. Additionally, COVID control measures such as lockdowns, restrictions on travel and movement have been associated with missed appointments among TB patients in the region and may have equally created challenges or deterrence among TB patients in Burkina Faso [22] Further qualitative research is required to better understand the barriers and contributors to patient delays in the time of COVID to inform appropriate responses.

The COVID pandemic has also decreased the capacity of health systems to provide services, with many health centers’ operations reduced to a minimum or even stopped during the pandemic in Burkina Faso and elsewhere, resulting in a significant reduction in patients’ access to TB care [23]. A study from Malawi found that factors such as limited PPE and fear of COVID exposure among staff contributed to reduced attendance of health care staff at TB facilities, including laboratory staff who were uncomfortable processing sputum samples due to fear of COVID-19 exposure which created delays in TB diagnosis [24]. However, in our study, we found that the overall diagnostic delay and system delay—time between first visit and diagnosis, and initiation of anti-TB treatment respectively—was reduced by more than half during the COVID period. There are several potential explanations for this finding. First, the decreased demand on the health system—as evidenced by the increasing time between symptom onset and attendance at a health facility—may have resulted in a smaller workload for TB staff. Second, access to GeneXpert machines increased during the COVID pandemic in Burkina Faso which facilitated rapid diagnosis of TB. However, we also noticed a greater proportion of TB clinically diagnosed during the COVID-19 period, which may be in response to access barriers among patients coming for TB diagnosis who may have been presenting more clinical symptoms due to suffering from more advanced TB. Finally, the similarity among symptoms of COVID and TB may have enhanced routine screening for TB as part of a differential diagnosis in the context of COVID [25].

This study also identified changes to the treatment pathways taken by TB patients during the COVID-19 pandemic. Specifically, this study found that the use of public health services as the first point of contact with the health system decreased while the first use of private health centers increased during COVID, compared to the pre-COVID period. This could be explained on one hand by an improvement in the supply of private health care services and providers, and on the other hand by a drop-in activity in the provision of general public health services due to COVID-19 in Burkina Faso, which led to significant absences of public employees due to COVID-19. This shift of patients to the private sector could also be explained by the fact that patients preferred to avoid the public services which acted as the first line services for suspected COVID-19 patients.

Our study carries some limitations. Our data collection relied on patient recall which could be a source of memory or responder bias. While we attempted to identify the changes in health seeking behavior and access to TB services before and during the COVID pandemic, we cannot demonstrate causality and acknowledge that there may be additional explanations behind the changes reported here. Further qualitative research is required to better understand patient experiences during the pandemic period in Burkina Faso.

Nonetheless, this is the first study to our knowledge to compare health seeking behavior and practices during COVID to pre-pandemic times and the findings have important implications for health promotion efforts to ensure that health seeking and access to services remains unimpeded during times of emergency. Our findings suggest clear priorities for the national TB programme in Burkina Faso, including community-based sensitization work to overcome sense of fear among TB patients which interferes with health seeking behaviors and consideration for community-based TB screening and treatment provision interventions to overcome barriers such as travel restrictions or fear to TB service access. Collectively, these strategies can help ensure the continuation of crucial TB services during future public health emergencies to continue efforts towards End TB goals in Burkina Faso.

## Figures and Tables

**Figure 1 tropicalmed-07-00237-f001:**
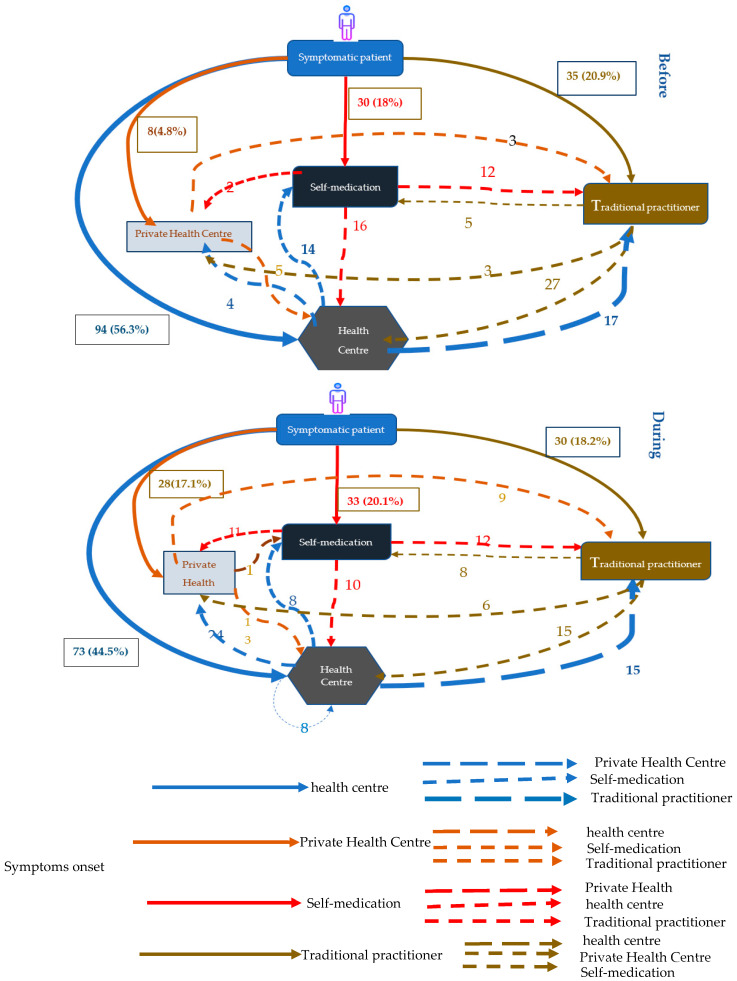
Therapeutic itinerary of patients Before and During COVID-19.

**Table 1 tropicalmed-07-00237-t001:** Sociodemographic characteristics of recruited TB patients during the ‘before’ and ‘during’ COVID periods.

Characteristics		Total	Before COVID n (%)	During COVID n (%)	*p*
Total number of participants		331	167	164	
Sex	Male	247	118 (47.8)	129 (52.2)	0.094
	Female	84	49 (58.3)	35 (41.7)	
Age (in years)	Mean (SD)	38.7	37.8 (15.3)	39.6 (14.4)	0.272
Age groups	0–14 years	6	4 (66.7)	2 (33.3)	0.335
	15–24 years	41	19 (46.3)	22 (53.7)	
	25–34 years	101	60 (59.4)	41 (40.6)	
	35–44 years	87	38 (43.7)	49 (56.3)	
	45–54 years	47	24 (51.1)	23 (48.9)	
	55–64 years	25	10 (40.0)	15 (60.0)	
	65 years and over	24	12 (50.0)	12 (50.0)	
Profession	Farmer/livestock breeder	120	71 (59.2)	49 (40.8)	<0.001
	Merchant	27	4 (14.8)	23 (85.2)	
	Housewife	56	36 (64.3)	20 (35.7)	
	Goldsmith	39	25 (64.1)	14 (35.9)	
	Other *	20	7 (35.0)	13 (65.0)	
Marital status	Bachelor	96	41 (42.7)	55 (57.3)	<0.001
	married	194	119 (61.3)	75 (38.7)	
	cohabitation	21	0(0.0)	21 (100)	
	Other **	20	7 (35.0)	13 (65.0)	
Region	Central East	189	91 (48.1)	98 (51.9)	0.392
	North Central	142	76 (53.5)	66 (46.5)	

* Student, unemployed, Public-Sector Employee, Private sector employee, or driver. ** divorced, Widowed or separated.

**Table 2 tropicalmed-07-00237-t002:** Clinical comorbidities and characteristics of recruited TB patients during the ‘before’ and ‘during’ COVID periods.

Characteristics		Total (n = 331)	Before COVID (n = 167)	During COVID (n = 164)	*p*
HIV status	Positive	19	8 (42.1)	11 (57.9)	0.464
	Negative	301	155 (51.5)	146 (48.5)	
	Unknown	11	4 (36.4)	7 (63.6)	
Body mass index	Mean (SD)	18.3	17.7 (2.9)	19.0 (3.3)	<0.001
BMI status	Normal (BMI ≥ 8.5– < 25)	138	58 (42.0)	80 (58.0)	0.001
	Underweight (BMI < 18.5)	186	109 (58.6)	77 (41.4)	
	Overweight (BMI ≥ 25– < 30)	6		6 (100.0)	
	Moderate obesity (BMI ≥ 30– < 35)	1		1 (100.0)	
Smoking	Yes	112	57 (50.9)	55 (49.1)	0.971
	No	219	110 (50.2)	109 (49.8)	
Alcohol consummation habits	Yes	87	47 (54.0)	40 (46.0)	0.515
	No	244	120 (49.2)	124 (50.8)	
Diabetes	Yes	4	3 (75.0)	1 (25.0)	0.628
	No	327	164 (50.2)	163 (49.8)	
Cough	Yes	302	160 (53.0)	142 (47.0)	0.003
	No	28	6 (21.4)	22 (78.6)	
Fever	Yes	264	127(77.0)	137(83.0)	0.215
	No	66	38(23.0)	28(17.0)	
Sweats	Yes	112	46 (41.1)	66 (40.2)	0.030
	No	215	117 (54.4)	98(45.6)	
Fatigue	Yes	211	98 (46.4)	113 (53.6)	0.107
	No	117	66 (56.4)	51 (43.6)	
Lack of appetite	Yes	206	103 (50.0)	103 (50.0)	
	No	121	60 (49.6)	61 (50.4)	
Pallor	Yes	18	6 (33.3)	12 (66.7)	0.252
	No	305	153 (50.2)	152 (49.8)	
Weight loss	Yes	255	124 (48.6)	131 (51.4)	0.486
	No	72	39 (54.2)	33 (45.8)	
TB diagnosis type	Clinically diagnosed	80	28 (35.0)	52 (65.0)	0.003
	Bacteriologically confirmed	251	139 (55.4)	112 (44.6)	

**Table 3 tropicalmed-07-00237-t003:** Median length of delays (in days) across the ‘before’ and ‘during’ COVID periods.

Delay Category	Overall Median Length of Delay in Days (IQR)	Before COVID Length of Delay in Days (IQR)	During COVID Length of Delays in Days (IQR)	*p*
overall delay	67.0 (33.0–128.0)	90.5 (43.8–164.5)	55.0 (30.0–91.0)	<0.001
patient delay	31.0 (15.0–68.0)	26.0 (10.0–43.0)	45.0 (26.0–90.0)	<0.001
system delay	33.0 (12.0–84.0)	44.0 (20.8–101.2)	20.0 (8.0–55.0)	<0.001
diagnostic delay	29.5 (8.0–80.2)	40.0 (17.0–97.0)	17.0 (5.0–52.0)	<0.001
processing time	2.0 (1.0–3.0)	1.0 (1.0–4.0)	2.0 (2.0–2.0)	0.003

## Data Availability

Requests for original datasets used in this manuscript can be directed to the corresponding author.

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
