# Peer review of "Delays in TB Diagnosis and Treatment Initiation in Burkina Faso during the COVID-19 Pandemic"

_tropicalmed, 2022, doi:10.3390/tropicalmed7090237_

Round 1

Reviewer 1 Report

Diallo et al present an interesting analysis on the effect of COVID-19 disruptions on the TB cascade of care in Burkina Faso. 

The manuscript is well presented and is an important and relevant topic for TB Global Health researchers and programme officers in general. There are methodological issues with the current version. This is a valuable work and I hope the authors revise this work to make these results available. My points below: 

Study design:

It is not clear what type of study this is, from the abstract or the methods. It says Before and After study, but that is not an actual study design and that makes the reader believe that this is sequential measure cohort study, where the same population is followed over time and measured for some endpoint at different stages. That is not the case. To me, it looks as if these are two cross-sectional studies compared.  This should be stated and clarified. 

Sample

line 128: says a combined sample size of 600 was set, but later in line 149 it says a total of 331. Which is?

Proportion comparisons

In Table 1. the percentages for each stratum are calculated across (adding the numbers "before" and "during"), but the more useful way of presenting them is over (e.g, what percentage of the "Before" group was 0 to 14 years old?) .

Similarly,  in Table 2. More importantly,  if the p values estimated in these tables were calculated using the presented proportions, then the results are invalid: For example, Table 1 says that the "Before" group had 118 males while the "During" group had 129. The proportions cited are 47.8 and 52.2 respectively. Is the p value of 0.094 calculated for the proportion on the table or for the correct one which is  118/167=70% against 129/164=78%?

Proportions in tables must be ammended.

Discussion

It is is interesting that the overall delay has decreased, principally at the account of the system and diagnostic delays. The reasoning here proposed is that there might be some hidden efficiencies , but I wonder if the fact that more people were diagnosed and treated in the private sector has an impact? The question is, when you recorded data, treatment and diagnosis also accounts for those performed in the private sector? This would be a very important point

Figure 1 is interesting but could be improved for clarity. The labels are not clear and a caption explaining what we see is necessary. 

Author Response

Thank you for reviewing this document and helping to improve it;

Reviewer 2 Report

It is a pleasure to review the manuscript entitled ‘Delays in TB diagnosis and treatment initiation in Burkina Faso during the COVID-19 pandemic’ written by Diallo et al.

The paper is relevant because it assesses the disruptions of TB detection services caused by the COVID-19 pandemic in a low-resource setting.  

However, the authors need to improve on the English as many sentences are poorly written. Again, there are major gaps in the methods applied for this study. A notable example is the discrepancy in the sample size indicated in the method section and the number of participants recruited in the study.  In line 125 to 128, the authors stated that ‘a sample size of approximately 320 was set for the ‘during’ COVID phase to match the sample size of the original study, giving a combined sample size of approximately 600’. There are problems with this sample size calculation; there are no documented statistical basis for its calculation and the figures do not match up. Additionally, there are ambiguity with some statements-the sample size of the original study. What is this original study?

Specific comments

Abstracts

·      Line 15: ‘Data was in two phases. The authors should add ‘collected’ or other words to complete the sentence.

·      Line 18: I suggest you start sentences with words rather than numbers

·      Line 19 and 20: there is no baseline data that can be used to compare the findings of 2018 (pre-covid-19). Therefore, the use of the word ‘increase’ is not appropriate. I recommend you use the results of 2018 as a baseline to compare with the findings of 2019.

·      The authors should add 19 to the spelling ‘COVID’

Methods

·      The authors should state whether the study was a retrospective or prospective

·      The authors used a cross-sectional study design to collect data in two phases: Phase 1 (December 2017- 15 March 2018) and Phase 2 (October-December 2020). There are challenges with this method

1.     Seasonal variation could be a challenge for this study. The delay in the presentation of patients in months of October, November and December could be different from other months of the year. What the authors could have done is to take information collected in the same months in the pre-COVID-19 and intra-COVID-19 era. This will help them circumvent the seasonality of presentation to hospitals and service provision.

2.     Again, the authors did not proffer any reason for selecting late 2017 and early 2018. In my opinion, what could have been appropriate is for the authors to select the year immediately following the COVID-19 pandemic i.e, 2019). There could have been changes overtime which may have influenced late presentation to TB care services.

·      In defining the reasons for maternal deaths, some authors use the three delays (delay in taking decision to seek medical care, delay in reaching health facilities and delay in receiving care after reaching health facilities) to define the causes of maternal mortality. This type of classification is more appropriate for planning and implementation of public health interventions and can be replicated in TB care. The authors can think about replicating this approach if feasible.

·      Line 119 (Participants and sample size calculations): It was not clear how the sample size was determined. Was a formula used to calculate the sample size or is that the sample was determined by convenience?

·      There appears to be discrepancy between the eligibility criteria for participants stated in the paper as against the actual participants recruited in the study. It appears information was collected from patients with presumed TB but the eligibility criteria state that patients were recruited only if they have bacteriologically or clinically confirmed TB. If you assessed patients with symptoms of TB until a diagnosis is made, then it applies patients with presumed TB were included in this study. This should be clarified.

Results

·      Line 126 to 129: The authors documented a sample size of 600 (320 each for the pre-and-intra-COVID-9 eras) but the result section stated a sample of 331 (164 each for phase 1 and 2). I wonder what could be the explanation for this difference.

·      Table 1: The authors should use more standardized English terminology (male, female, married, single etc.)

·      Line 163: A normal BMI has a value of 18.5 -24.9kg/m2 or whatever reference that is used. But it is unusual for the authors to categorize all patients with a BMI <25 Kg/m2 as underweight.

·      The format of the result presentation should be improved

·      It is difficult to understand the denominators for calculating the proportions of patients with symptoms and other variables in Table 2.

Discussion

·      It will be nice for the authors to discuss the result in the context of the healthcare infrastructure in Burkina Faso including distance between healthcare facilities and the road network.

·      It could also be nice for the authors to contextualize discussion in relation to practices such as traditional practices and seeking healthcare services in pharmacies

·      The authors surmise that they assess the health seeking behavior for patients with presumed or confirmed TB. In my opinion, this is inappropriate. The authors only assessed delays in accessing TB services but not potential factors that cause these delays (which may in fact include behavioral issues).

Author Response

Thank you for reviewing this document and helping to improve it

Round 2

Reviewer 2 Report

No further comments